# Identification of Hub Genes Correlated with the Initiation and Progression of CKD in the Unilateral Ureteral Obstruction Model

**DOI:** 10.3390/biomedicines13061316

**Published:** 2025-05-27

**Authors:** Xinxin Li, Junjie Li, Xiaobing Yao, Jun Yang

**Affiliations:** 1Department of Urology Surgery, Tongren Hospital Affiliated to Wuhan University (The Third Hospital of Wuhan), Wuhan 430060, China; 2Renmin Hospital of Wuhan University, Wuhan 430060, China

**Keywords:** chronic kidney disease, UUO, bioinformatics, hub genes

## Abstract

**Background:** Chronic kidney disease (CKD) is a global health problem marked by a persistent deterioration in the function of the nephrons and kidneys. To identify novel therapies for CKD, we investigated the molecular targets associated with the initiation and progression of the disease. **Methods:** The transcriptional profile dataset of GSE42303 was downloaded from GEO (The Gene Expression Omnibus). Utilizing the R package limma, the differentially expressed genes (DEGs) were identified between control (Con) and unilateral ureteral obstruction (UUO) mice. Then, functional enrichment, protein–protein interactions (PPI) and subsequent hub genes were identified by multiple bioinformatics approaches. Further validations of these hub genes were confirmed through the GSE118339 dataset and in vivo experiments. **Results:** We found 381 DEGs between Con and UUO mice (308 up-regulated genes and 73 down-regulated genes). GO functions and pathway analysis indicated that DEGs were mainly enriched in activities associated with inflammation and fibrosis. The mRNA expressions of nine hub genes were identified and confirmed by dataset GSE118339 and in vivo experiments. **Conclusions:** The hub genes Fgg, Penk, Ckap4, and Gpc3 may be new prospective targets for the treatment of the initiation and progression of CKD.

## 1. Introduction

Chronic kidney disease (CKD) is a progressive, incurable illness with substantial morbidity and fatality rates [1]. Parenchymal cell loss, chronic inflammation, fibrosis, and diminished renal regeneration capacity contribute to the pathogenesis of CKD [2], which could advance to end-stage renal disease (ESRD). However, our comprehension of the molecular pathways underlying the progression of CKD is limited. Moreover, the current therapeutic effects are restricted and can only be used to delay disease progression [3]. In the initial phases of CKD, it is essential to recognize relevant molecular markers or prospective molecular targets for medication and intervention to treat CKD and reduce its development.

Renal inflammation is crucial for the onset and development of CKD. Several types of kidney damage can induce renal inflammation as a defensive reaction [4,5]. In the initial stages of kidney injury, the damaged kidney cells produce a variety of inflammatory factors, which can be chemotactic and recruit various inflammatory cells [4,6]. The recruited and activated inflammatory cells can produce inflammatory factors that further aggravate the injury, and they can secrete cytokines and growth factors that promote the conversion of interstitial cells and renal tubular cells into fibroblasts, which in turn produce extracellular matrix and promote the formation of renal fibrosis [7,8]. For example, MCP-1 is a major chemokine released by damaged renal tubular epithelial cells and induces T cell and monocyte infiltration, which in turn produces several cytokines to promote the progression of renal fibrosis [9].

In the present study, transcriptome profiles GSE42303 were employed to investigate possible biotargets or biomarkers. Ten hub genes were identified, including Fgg, Penk, Ckap4, Gpc3, etc. We further confirmed the findings by comparing the degree of these genes’ expression in vivo. These four hub genes might be possible targets or biomarkers in the onset and progression of CKD.

## 2. Materials and Methods

### 2.1. Affymetrix Microarray Data Information

GSE42303, GSE118339, and GSE30528 gene expression profile datasets came from the GEO database at the NCBI (https://www.ncbi.nlm.nih.gov/geo/, accessed on 2 December 2021). The GSE42303 dataset, which is based on the GPL8321 platform (Affymetrix Mouse Genome 430A 2.0 Array; Affymetrix, Santa Clara, CA, USA), contains three Sham samples and three UUO samples. The GSE118339 dataset, based on the GPL19057 platform (Illumina NextSeq 500; Illumina, San Diego, CA, USA), has three Sham and four UUO samples, which were used as an external validation dataset to estimate hub gene expression levels. The GSE30528 dataset is based on the GPL571 platform (Affymetrix Human Genome U133A 2.0 Array; Affymetrix, Santa Clara, CA, USA). It contains nine control samples and thirteen diabetic nephropathy samples, which are used to verify the expression of core genes in diabetic nephropathy models.

### 2.2. Identification of DEGs

Firstly, we normalized the matrix data and calculated the average value of numerous probes targeted at a particular gene. Then, by comparing UUO with Sham samples, we identified DEGs using the R package limma. The threshold for DEGs was |log2 Fold Change (FC)| > 2 and *p*-values < 0.05.

### 2.3. Analysis of DEGs’ Functional Enrichment

DAVID (https://david.ncifcrf.gov/, accessed on 2 December 2021) is a database used for functional enrichment analysis to assess the biological importance of potential DEGs. DAVID evaluated the DEGs for the Kyoto Encyclopedia of Genes and Genomes (KEGG) pathways analysis and enriched GO analysis (BP: biological process, CC: cellular component, MF: molecular function).

### 2.4. Protein–Protein Interaction (PPI) Network, Module Analyses, and Hub Genes Identification

We used the Search Tool for the Retrieval of Interacting Genes (STRING) database (https://string-db.org/, accessed on 2 December 2021) to predict the PPI network of DEGs. The DEGs were uploaded to the STRING, and interaction scores were set at 0.7. The plug-in APP MCODE in Cytoscape (3.8.1) was used to identify key modules and hub genes within this PPI network. The threshold setting for the MCODE scores was higher than 10 and nodes greater than 19.

### 2.5. Animal Model

To confirm the level of central gene mRNA expression in vivo, weight-matched male mice aged 8 to 12 weeks were utilized in this research. The mice were bred in a sterile environment devoid of pathogens at the Wuhan University Animal Experimentation Center. The approval for all animal studies was granted by the Wuhan University Animal Ethics Review Board and conducted under the standards of the National Health and Medical Research Council of China (#SY2021-019).

To create the UUO model, we sedated mice with 30% isoflurane, exposed their left ureters, and ligated them with silk sutures. Sham-operated animals received the same surgery without obstructing the ureter. Mice were kept breeding for 3, 7, and 14 days after surgery and sacrificed for further experiments.

### 2.6. Morphometric Analysis

Kidneys were preserved in 4% paraformaldehyde and paraffin-embedded. Following standard protocol, Sirius red and Masson’s Trichrome stains were applied to 4 µm-thick kidney sections. The slides were viewed using a microscope (Olympus, Tokyo, Japan) and analyzed using Image J (1.44).

### 2.7. Immunofluorescence Staining

Kidney slides that were paraffin-embedded and deparaffinized measured 4 µm in thickness. These slides were made using a 10% normal donkey serum (Vector Labs, Burlington, ON, Canada) solution diluted in PBS. Following exposure to a-SMA, FN, and F4/80 primary antibodies, Alexa Flour 488-conjugated secondary antibodies were applied to the slides. DAPI was used to stain the nuclear (Life Technologies, Carlsbad, CA, USA). The slides were observed by a microscope (Olympus, Tokyo, Japan). For each group, 10 fields were randomly selected from five mice. The positive area % was measured using Image J at a magnification ×200 (https://ij.imjoy.io/; National Institutes of Health, Bethesda, MD, USA).

### 2.8. Real-Time qPCR

A total of 10 mg of mouse kidney samples was subjected to RNA isolation using a Trizol reagent. The recovered RNA was then converted into cDNA using the SweScript RT II First Strand cDNA Synthesis Kit (Servicebio, Wuhan, China). The qRT-PCR experiment was conducted using a Bio-Rad Cycler instrument (CFX Connect, Bio-Rad, Hercules, CA, USA). The primers specific to the sequence are provided in Table 1.

### 2.9. Western Blot

Tissue lysates were subjected to western blotting. Concisely, the entire protein lysates were placed onto the SDS-PAGE gel and then moved to a PVDF membrane (Merch Millipore, Darmstadt, Germany). Following a 1 h incubation at room temperature in a TBS solution containing 5% skim milk, the membrane was then incubated overnight at 4 °C with primary antibodies for a-SMA, FN, IL-6, and GAPDH. This was followed by an additional 1 h incubation at room temperature with secondary antibodies. A chemiluminescence detection system manufactured by Bio-Rad Laboratories was utilized for the purpose to detect protein bands. Integrated density values of bands were evaluated using Image J Software.

### 2.10. GeneMANIA

GeneMANIA is a versatile database that offers information on gene function, protein–protein interactions, gene-dataset linkages, functionally similar genes, and shared properties among related genes. You may access it at http://www.genemania.org (accessed on 10 December 2021) [10].

### 2.11. Metascape

Metascape (https://metascape.org, accessed on 12 December 2021) is a dependable internet source for the annotation and enrichment analysis of genes [11]. Within this study, the GO and KEGG enrichment functions of hub genes were examined in Metascape.

### 2.12. Statistics Analysis

Using GraphPad Prism 8 (San Diego, CA, USA), the data were analyzed and given as mean ± standard error of the mean (SEM). Intergroup comparisons were conducted using unpaired 2-tailed *t*-test and one-way ANOVA, while post-comparisons were conducted using the Tukey test. *p* < 0.05 was considered to be statistical significance.

## 3. Results

### 3.1. DESs Identification

Using the gene expression profile of the GSE42303 dataset, we have found a total of 381 DEGs, consisting of 308 genes that are up-regulated and 73 genes that are down-regulated, comparing UUO samples to Sham samples, shown in a volcano plot (Figure 1A). Figure 1B displays a heatmap that illustrates the profile of the DEGs.

### 3.2. Enrichment Analyses

To get an additional understanding of the critical potential pathways and functional enrichment of DEGs in the CKD progression caused by UUO, the biological process (BP) of GO term, KEGG pathway, and Reactome pathway for the DEGs were conducted. Based on the results of DAVID, innate immune response, inflammatory response, immunological system, and other processes were among the processes in which the BP revealed that the DEGs were considerably enriched (Figure 2A). For KEGG pathways analysis, the most enriched pathways were phagosome, complement and coagulation cascades, ECM-receptor interaction, etc. (Figure 2B). The examination of Reactome pathway enrichment showed that the DEGs were primarily associated with integrin cell surface contacts, ECM proteoglycans, collagen and extracellular matrix breakdown, etc. (Figure 2C). In summary, the DEGs mainly were associated with inflammation and fibrotic response in the development of CKD.

### 3.3. Building PPI Networks and Module Analysis

To study the associations among these DEGs, a PPI network was constructed using the database STRING. A total of 377 nodes and 1179 edges constituted the PPI network (Appendix A). Using APP MCODE, we then generated 11 modules based on the PPI network; the top two key modules with scores > 10 were identified (Figure 3).

### 3.4. Identification of Hub Genes

Moreover, using the Cytoscape plug-in APP Cytohubba, the hub genes were screened using the MCC approach (Figure 4A). The genes with the highest score, ranked in the top 10, were identified as hub genes, including *Fgg*, *Fn1*, *Timp1*, *C3*, *Penk*, *Ckap4*, *Trf*, *Gpc3*, *Apoe*, *and Fbn1*. The detailed information about hub genes was described in Table 2. Then, these 10 hub genes’ mRNA expression levels were confirmed using the GSE118339 dataset. The mRNA levels of hub genes, with the exception of Trf, were significantly higher in the UUO group (Figure 4B–J). Due to Trf not being identified in dataset GSE118339, the expression result of Trf was not shown in Figure 4.

### 3.5. Validation of Hub Genes In Vivo

To validate the relevant hub genes in vivo, we attempted to induce a mouse model of renal fibrosis with UUO (Figure 5A). Inflammation and collagen deposition become more severe with increasing occlusion time in the UUO group (Figure 5B–G). The level of fibrosis increased over time during the surgery, as shown by the results from Sirius red staining, Masson staining, a-SMA, FN, and F4/80’s immunofluorescence staining (Figure 5B–G). After UUO surgery, the mRNA and protein expression levels and inflammatory biomarkers, a-SMA, FN, IL-6 and MCP-1, were up-regulated (Figure 6A,B). These results demonstrated that the UUO mice were effectively established. We further confirmed the degree of hub gene expression, including *Fgg*, *Fn1*, *Timp1*, *C3*, *Penk*, *Ckap4*, *Trf*, *Gpc3*, *Apoe*, *and Fbn1*, and observed that the degrees of mRNA expression of *Fgg*, *C3*, *and Gpc3* were significantly decreased with the treatment time but still higher than those in the sham mice, while the degrees of mRNA expression of the other seven hub genes were significantly increased (Figure 6C). The findings confirmed that these 10 hub genes may be applicable as possible treatment targets. In addition, we verified the expression of hub genes in the diabetic nephropathy model, and we found that the expression of hub genes was consistent with our expectations (Appendix A).

### 3.6. Predict the Co-Expressed Genes of Hub Genes and Their Enrichment Functions

GeneMANIA was used to find co-expressed genes of hub genes and predict their interactions (Figure 7A). The interactions of genes were mainly enriched in pathways and processes for extracellular matrix organization, degradation of the extracellular matrix, cell junction organization, and inflammatory response, which are associated with fibrosis and inflammation (Figure 7B). The DisGeNET function in Metascape revealed that these genes are related to CKD stage 5, renal insufficiency, and acute kidney injury (Figure 7C), which confirmed that a significant part of the development of CKD is played by these 10 hub genes.

## 4. Discussion

Although several biomarkers of CKD have been identified, including urinary N-acetyl-D-aminoglycosides (NAG), neutrophil gelatinase-associated lipid transport protein (NGAL), and kidney injury molecule-1 (KIM-1) [12,13], it is still necessary to find effective targets for alleviating or treating the progression of CKD. In the current investigation, we determined the potential biomarkers in CKD through multiple bioinformatics analyses of two datasets. Through bioinformatics analyses of GSE42303 and hub genes validation of GSE118339, we obtained 10 hub genes. Among them, the functions of Fn, Timp1, C3, Apoe, Trf, and Fbn1 were already previously thoroughly studied in renal fibrosis. Therefore, we discussed the function of Fgg, Penk, Ckap4, and Gpc3, which were verified in vivo and identified as potential molecular biomarkers in CKD.

The protein encoded by the Fgg gene is the subunit of fibrinogen, a glycoprotein found in the blood consisting of two groups of α and polypeptide chains. Both high and lowered fibrinogen levels are clinically significant. Fibrinogen plays a vital function in coagulation. Fibrinogen is essential for maintaining hemostasis, and there is now mounting evidence that plasma fibrinogen also plays an important role in encouraging the progression of inflammation, fibrosis, and cancer. Fgg is elevated in mice with pulmonary fibrosis, and it may be used as a biomarker candidate for pulmonary fibrosis [14]. Fgg could be a prognostic key gene in renal clear cell carcinoma, and its expression could predict response to sunitinib. It may Facilitate the advancement of novel therapies for this ailment [15]. The elevated levels of Fgg have been detected in the protein urine of patients with acute renal transplant rejection. Fgg separates acute rejection from BK viral nephritis and it is helpful for accurately and delicately diagnosing acute rejection of a kidney transplant without the need for invasive procedures [16]. In renal fibrosis, fibrinogen may directly promote renal fibroblast proliferation in a dose-dependent manner and synergistically with TGF-β1 to increase fibroblast proliferation and activate the TGF-β1/PSMAD2 signaling pathway [17]. Roland et al. employed fibrinogen knockout mice to create a UUO model and discovered that, in comparison to homozygous fibrinogen-deficient mice, heterozygous mice exhibited more pronounced renal interstitial fibrosis and substantial fibrinogen accumulation in the renal interstitium [18]. Wang et al. identified urinary fibrinogen as an independent risk factor for the development of CKD to ESRD in their research, which included several CKD types [19,20]. Fgg is associated with renal interstitial fibrosis and can be a potential biomarker for chronic kidney disease [21].

The Penk-encoded preproprotein undergoes proteolytic processing to generate several protein products. The cleavage products may possess many biological activities [22]. Increased Penk levels correlate with glomerular and tubular dysfunction and have recently been recognized as a potential biomarker for renal function, offering superior detection of renal damage compared to creatinine-based tests. Plasma proenkephalin levels correlate with glomerular filtration rate (GFR), as evaluated by established methodologies [23]. Prior research has shown its correlation with adverse clinical outcomes across several contexts, including sepsis, heart failure, renal transplantation, and CKD [24]. Moreover, in healthy people, elevated proenkephalin levels correlate with reduced GFR and an augmented risk of CKD [25]. Our findings indicate that Penk mRNA levels in mouse kidneys progressively elevated with the advancement of CKD, implying that Penk may serve as a dependable biomarker for CKD.

Cytoskeleton-associated protein 4 (Ckap4) is found in lipid droplets, nuclear specks, and rough endoplasmic reticulum, where it mediates the endoplasmic reticulum’s attachment to microtubules. Studies have shown that by affecting MMP2 synthesis and YAP phosphorylation, Ckap4 influences VC in CKD, hence accelerating the course of the disease [26]. Through activation of the PAR2/p38/JNK signaling pathway, Ckap4 is essential in mediating trypsin-like induced phenotypic alterations in atrial fibroblasts. Overexpression of Ckap4 significantly enhances cellular proliferation, migration, and the production of type I collagen and fibronectin. Additionally, it upregulates matrix metalloproteinase-1 (MMP-1) expression while simultaneously increasing tissue inhibitor of metalloproteinase-1 (Timp-1) levels [27]. A study employing OLINK plasma proteomics analysis on patients with varying stages of renal dysfunction revealed that individuals with CKD exhibited increased levels of SASP cytoskeleton-associated protein 4 and pentraxin-related protein 3 (PTX3), indicating a potential involvement of these proteins in CKD pathogenesis [28]. Ckap4 has a role in preserving endoplasmic reticulum sheets, is linked to the progression of CKD, and is a possible target for pharmacological interventions aimed at treating renal fibrosis. Ckap4 is related to inflammation and fibrosis in our research, and its expression is considerably elevated in a mouse model of UUO. It is necessary to further investigate its molecular processes in CKD and renal fibrosis.

Glypican-3 (Gpc3) is a cell-surface proteoglycan containing heparin sulfate. Gpc3 attaches to the cell surface via a GPI-anchor and negatively inhibits the Hedgehog signaling pathway through a competition for Hedgehog interaction with the Hedgehog receptor PTC1 [29]. Gpc3 binds to the Frizzled Wnt receptor to regulate the canonical Wnt signaling pathway and increasing its association with Wnt ligands [30]. By modulating cellular responses to BMP4, Gpc3 is involved in limb shape and skeletal development. Additionally, it regulates the impact of growth hormones BMP2, BMP7, and FGF7 on the formation of renal branching [31]. In biliary atresia liver, there was a favorable correlation between the expression of glypican-3 and the grade of hepatic fibrosis in biliary atresia. Furthermore, there was a correlation between elevated levels of glypican-3 expression and unfavorable survival outcomes, suggesting that liver overexpressed glypican-3 is associated with hepatic dysfunction and fibrosis [32]. The mRNA expression level of Gpc3 increases synchronously with the expression of fibrotic and inflammatory biomarkers in mice models of UUO. The molecular role of Gpc3 in renal inflammation and fibrosis should be studied further.

Although we have identified potential biomarkers in CKD through multiple bioinformatics analyses and successfully verified their expression in a mouse model of chronic kidney disease, our study still has flaws. First, we did not conduct a deeper mechanistic study by up-regulating and down-regulating the expression levels of hub genes. Second, the gene chip data we used contained a limited number of mouse samples. The chronic kidney disease model we used to screen hub genes was the UUO model of mice, and they were verified in the UUO model and the diabetic nephropathy model. Verification is also needed in other types of chronic kidney disease models. Finally, although mouse models are very useful in simulating certain specific stages of CKD pathology (such as fibrosis, inflammation, and decreased renal function), they cannot fully replicate the entire process of the chronic, complex, and heterogeneous development of human CKD.

## 5. Conclusions

We identified the hub genes of CKD (Fn, Timp1, C3, Apoe, Trf, Fbn1, FGG, Penk, Ckap4, and Gpc3) through multiple bioinformatics analyses and successfully verified their expression in a mouse chronic kidney disease model. Enrichment analysis showed that these genes were associated with chronic kidney disease, renal insufficiency, and acute kidney injury, and they are expected to become potential molecular biomarkers and therapeutic targets for CKD. More research is needed in the future to evaluate the role of these genes in renal fibrosis.

## Figures and Tables

**Figure 1 biomedicines-13-01316-f001:**
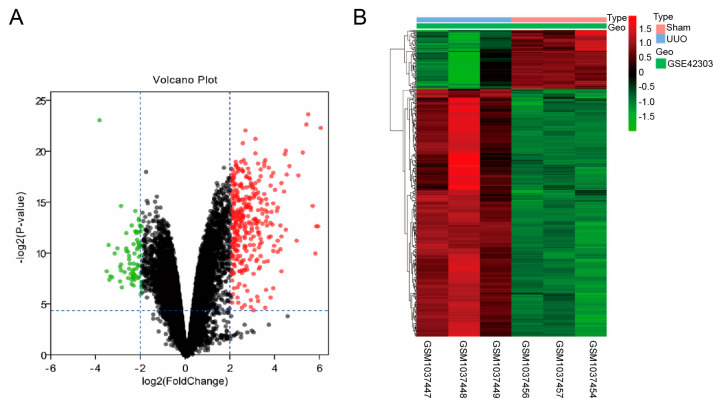
Identification of DEGs from the GSE42303 dataset. (**A**) The volcano plot visualizes DEGs. Red: up-regulated genes; Green: down-regulated genes (**B**) The expression profiles of DEGs are shown as the heatmap.

**Figure 2 biomedicines-13-01316-f002:**
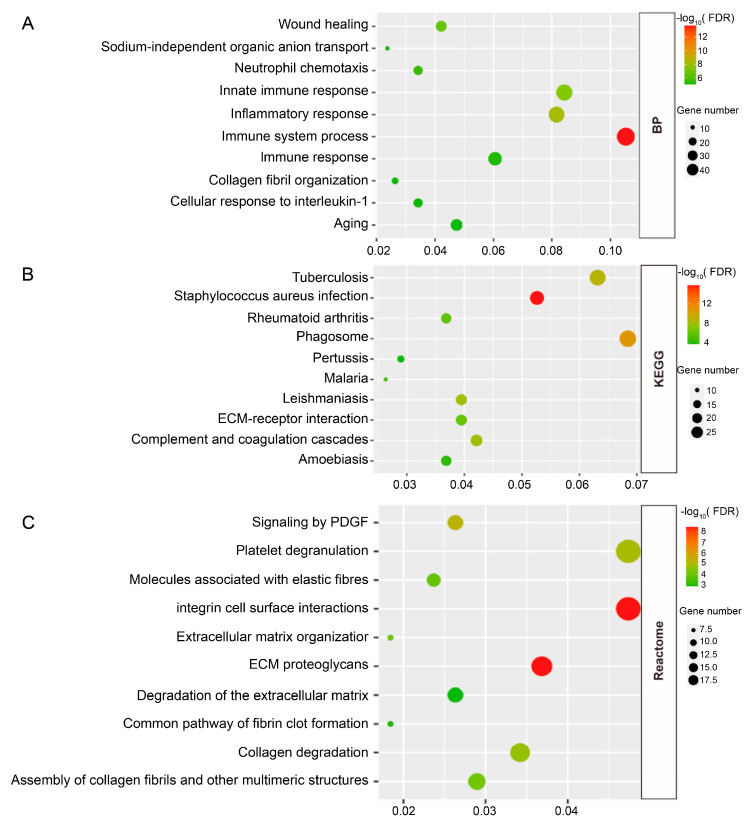
Functional enrichment analyses of DEGs. (**A**) The top 10 significantly enriched biological processes. (**B**) The top 10 significantly enriched KEGG pathways. (**C**) The top 10 significantly enriched Reactome pathways. *p*-value < 0.05 was set as the threshold.

**Figure 3 biomedicines-13-01316-f003:**
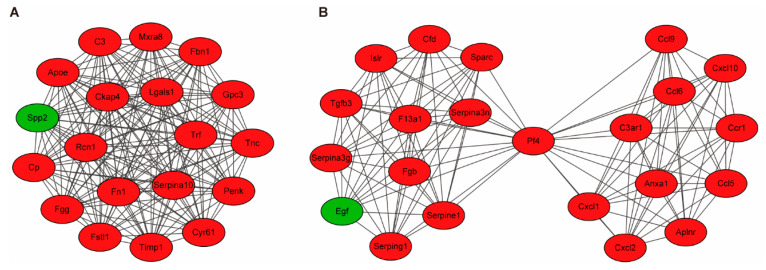
Top two identified significant modules analyses. (**A**,**B**) were the top two significant modules identified by MCODE from the PPI network. MCODE scores were 19 and 11, respectively. Red nodes indicated up-regulated genes, and green nodes represented down-regulated genes.

**Figure 4 biomedicines-13-01316-f004:**
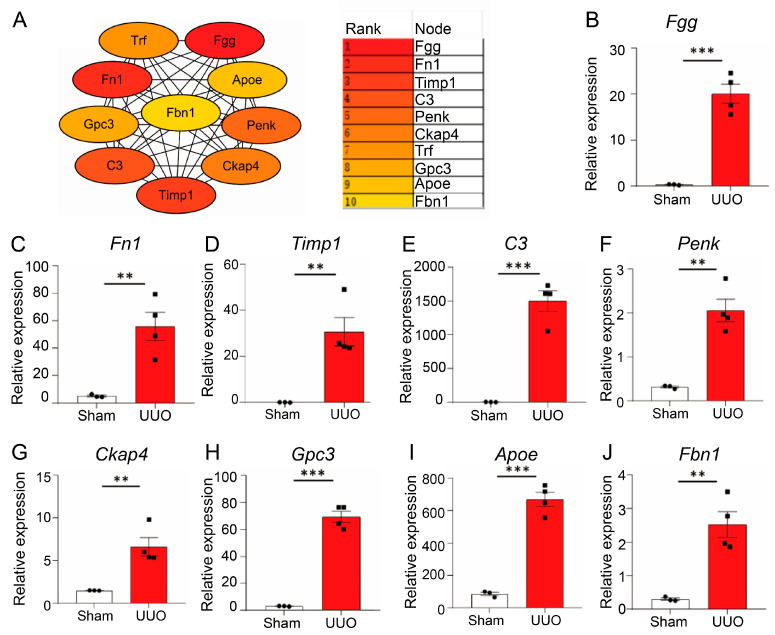
The expression level of hub genes was confirmed in GSE118339. (**A**) Ten hub genes were identified based on the MCC method using the plug-in APP Cytohubba in Cytoscape. (**B**–**J**) The mRNA levels of hub genes in GSE118339 are presented. ** *p* < 0.01, *** *p* < 0.001.

**Figure 5 biomedicines-13-01316-f005:**
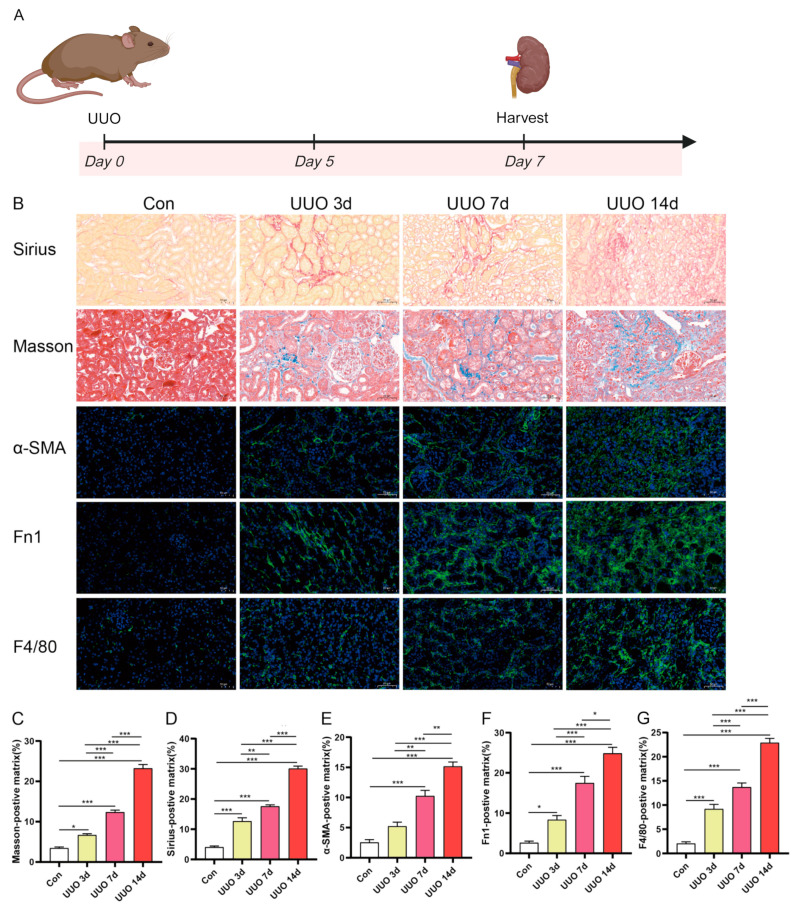
Inflammation and collagen deposition become more severe with increasing occlusion time in the UUO mice. (**A**) Scheme of the unilateral ureteral obstruction. (**B**) Representative micrographs of Sirius red, Masson staining, a-SMA, FN, and F4/80’s immunofluorescence staining in Sham and UUO mice. Scale bar = 50 µm. (**C**–**G**) Quantification analysis of Sirius red, Masson staining, a-SMA, FN, and F4/80’s immunofluorescence staining in control and UUO mice. * *p* < 0.05, ** *p* < 0.01, *** *p* < 0.001.

**Figure 6 biomedicines-13-01316-f006:**
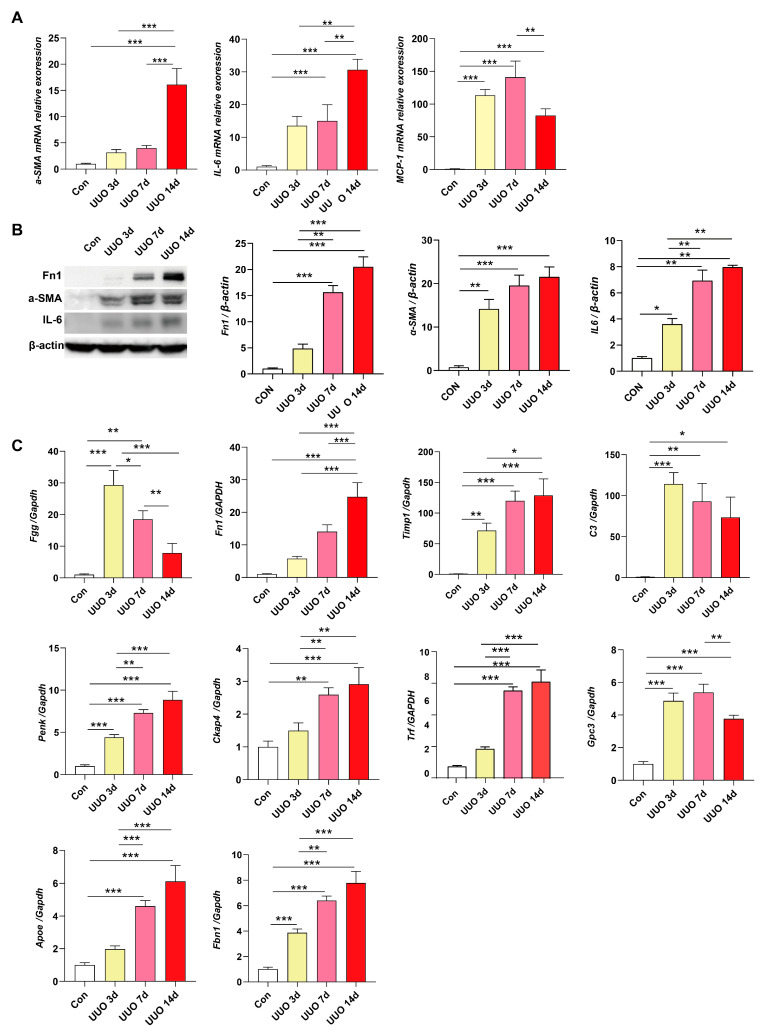
The expression levels of hub genes were confirmed in vivo. (**A**,**B**) Detection of fibrotic and inflammatory biomarkers with qRT-PCR and Western Blotting. (**C**) The expression of differentially expressed genes in vivo was detected by qRT-PCR. * *p* < 0.05, ** *p* < 0.01, *** *p* < 0.001.

**Figure 7 biomedicines-13-01316-f007:**
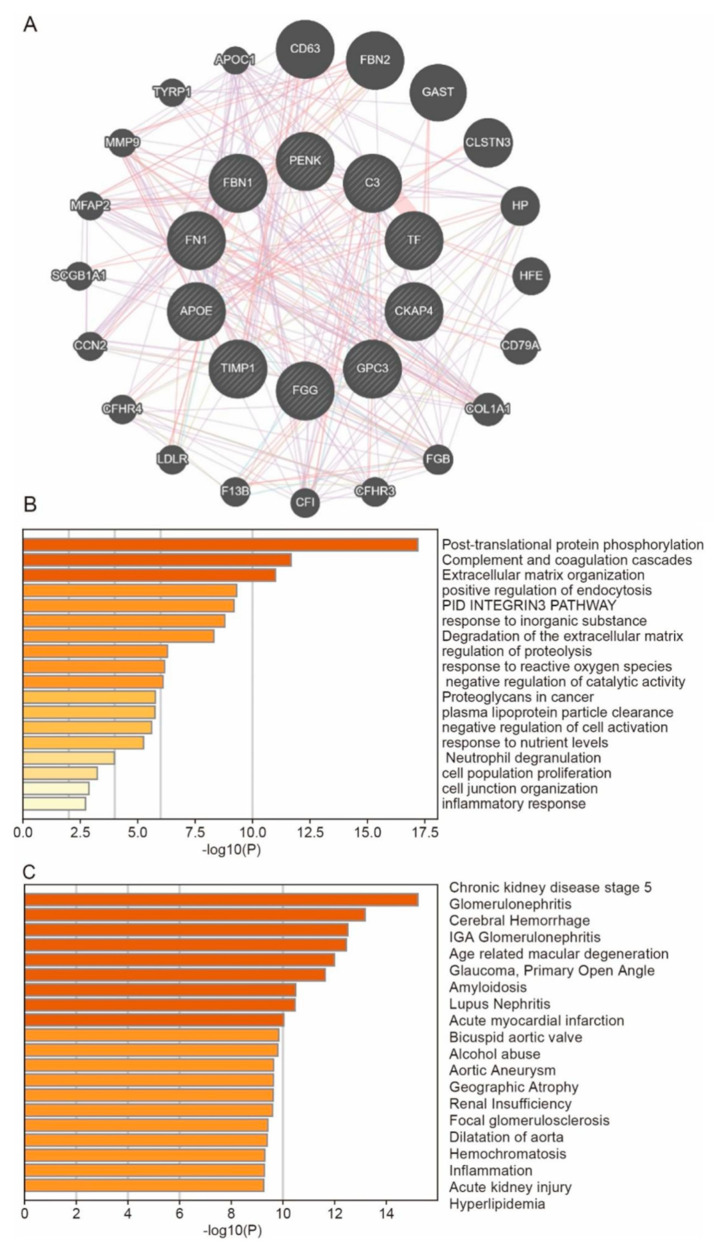
Enrichment of hub genes and their co-expressed genes. (**A**) The interaction relationship between hub genes and their co-expressed genes comes from geneMANIA. (**B**) Pathway and Process Enrichment Analysis from Metascape. (**C**) Summary of enrichment analysis in DisGeNET from Metascape.

**Table 1 biomedicines-13-01316-t001:** The primer sequences used in this study.

Gene	Forward Primer	Reverse Primer
*Fn1*	GGCCACCATIACTGGTCTGG	GGAAGGGTAACCAGTTGGGG
*α-SMA*	AGCCATCTTTCATIGGGATGG	CCCCTGACAGGACGTTGTTA
*Fgg*	GCACCACAGAGTTTTGGCTG	ATAGTCCGCAGTGCTGGTTC
*Timp1*	GCAACTCGGACCTGGTCATAA	CGCTGGTATAAGGTGGTCTCG
*C3*	TCCTTCACTATGGGACCAGC	TGGGAGTAATGATGGAATACATGG
*Penk*	AGGCGCGTTCTTCTCTCCTA	AGTGTGCACGCCAGGAAAT
*Ckap4*	GGCTGGTATGTCCATCACGTC	CTTGCAGGGATTGGACCTTCTG
*GPC3*	ATCCAGCCGAAGAAGGGAAC	TTCTTGTCCGTTCCAGCACA
*Apoe*	ATCCGATCCCCTGCTCAGAC	TGATIGGCCAGTCTCCCCTT
*Fbn1*	ACGATACTTGAAGAGGACAGGC	TGTCCTGATGCAGAGAGGTC
*IL-6*	CTCATICTGCTCTGGAGCCC	CAACTGGATGGAAGTCTCTTGC
*Trf*	GTGTGACGAGTGGAGCATCA	TCCGCTTCTCCGTTCACAAT
*MCP-1*	CACTCACCTGCTGCTACTCA	GCTTGGTGACAAAAACTACAGC
*GAPDH*	GGTTGTCTCCTGCGACTTCA	TGGTCCAGGGTTTCTTACTCC

**Table 2 biomedicines-13-01316-t002:** The function of the hub genes.

Gene	Function ^#^
*Fgg*	Fibrinogen gamma chain: Together with fibrinogen alpha (FGA) and fibrinogen beta (FGB), it polymerizes to form an insoluble fibrin matrix. It has a major function in hemostasis as one of the primary components of blood clots. In addition, it functions during the early stages of wound repair to stabilize the lesion and guide cell migration during re-epithelialization.
*Fn1*	Fibronectin 1: Fibronectin type III domain containing endogenous ligands.
*Timp1*	Metalloproteinase inhibitor 1: Metalloproteinase inhibitor that functions by forming one-to-one complexes with target metalloproteinases, such as collagenases, and irreversibly inactivates them by binding to their catalytic zinc cofactor. It acts on MMP1, MMP2, MMP3, MMP7, MMP8, MMP9, MMP10, MMP11, MMP12, MMP13, and MMP16. It does not act on MMP14. It also functions as a growth factor that regulates cell differentiation, migration, and cell death and activates cellular signaling cascades via CD63 and ITGB1. It plays a role in integrin signaling.
*C3*	Complement C3: C3 plays a central role in the activation of the complement system. Its processing by C3 convertase is the central reaction in both classical and alternative complement pathways. After activation C3b can bind covalently, via its reactive thioester, to cell surface carbohydrates or immune aggregates; C3 and PZP-like alpha-2-macroglobulin domain containing.
*Penk*	Proenkephalin-A: Met- and Leu-enkephalins compete with and mimic the effects of opiate drugs. They play a role in a number of physiologic functions, including pain perception and responses to stress.
*Ckap4*	Cytoskeleton-associated protein 4: High-affinity epithelial cell surface receptor for APF.
*Trf*	It is predicted to enable iron chaperone activity; iron ion binding activity; and transferrin receptor binding activity. It is involved in several processes, including ERK1 and ERK2 cascade; osteoclast differentiation; and positive regulation of bone resorption. It acts upstream of or within SMAD protein signal transduction.
*Gpc3*	Glypican-3: Cell surface proteoglycan that bears heparan sulfate. It inhibits the dipeptidyl peptidase activity of DPP4. It may be involved in the suppression/modulation of growth in the predominantly mesodermal tissues and organs. It may play a role in the modulation of IGF2 interactions with its receptor and thereby modulate its function. It may regulate growth and tumor predisposition.
*Apoe*	Apolipoprotein E: It mediates the binding, internalization, and catabolism of lipoprotein particles. It can serve as a ligand for the LDL (apo B/E) receptor and for the specific apo-E receptor (chylomicron remnant) of hepatic tissues.
*Fbn1*	Fibrillin-1: Structural component of the 10–12 nm diameter microfibrils of the extracellular matrix, which conveys both structural and regulatory properties to load-bearing connective tissues. Fibrillin-1 containing microfibrils provide long-term force bearing structural support.

^#^ The function of genes was obtained from the NCBI database (https://www.ncbi.nlm.nih.gov/ (accessed on 23 December 2021)).

## Data Availability

Any interested party is welcome to access the datasets used in this study by contacting the corresponding author.

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
