# Peer review of "Identification of Hub Genes Correlated with the Initiation and Progression of CKD in the Unilateral Ureteral Obstruction Model"

_biomedicines, 2025, doi:10.3390/biomedicines13061316_

Round 1
Reviewer 1 Report
Comments and Suggestions for Authors
To identify novel therapies for chronic kidney disease, the authors aimed to investigate the molecular targets associated with the initiation and progression of the disease.
I recommend that you correct the article from a technical editing point of view. You have many mistakes (for example Cakp4” instead of “Ckap4”,; repetition - “multiple bioinformatics informatics”).
I also recommend that you seek the help of a native speaker or an authorized translator.
The discussion mostly summarizes findings. I recommend you to discuss the mechanistic implications of the hub genes in CKD progression.
Limitations are not explicitly discussed (e.g., limited sample sizes, differences between mouse and human disease).
The article has 22 references, the last one from 2022. I recommend adding more recent references from recent years. This way, you will also expand the discussions.
However, this study is scientifically valuable and methodologically sound.
Reviewer 2 Report
Comments and Suggestions for Authors
Comments and Suggestions:
Title: Hub genes in chronic kidney disease progression.
Reviewer’s report:
The manuscript by Li et al., investigated the identification of novel therapies for chronic kidney disease (CKD) using data from GEO database. The authors have identified 381 DEGs between unilateral ureteral obstruction (UUO) mice and controls. Also, the GO functions and pathways showed the involvement of DEGs in inflammation and fibrosis. The nine hub-genes were validated in GEO dataset and in vivo experiments. They finally concluded that hub genes FGG, PENK, CAKP4, and GPC3 may be involved in the occurrence and progression of CKD.
The manuscript does not showed novelty but there are few points which need to be addressed.
Major Points:
- Line 64-65: The fold change cutoff of >2 is very stringent, due to which the identification of other important genes which are significantly deregulated might have been missed. Usually the fold change of >1 is used in most cases.
- Figure 1A: The volcano plot is usually plotted between log2FC and -log10(pvalue). But in the figure the authors have used -log2(p-value). If it mistakenly written, then also the cutoff will be 1.3 which will increase the number of identifications. Please clarify.
- The number of samples used for bioinformatics analysis is very less. Please mention the limitations of this study.
- Supplementary figure 2: The GSE30528 dataset is not mentioned in the materials and methods section. Why this additional validation data is needed when validation is done using a GEO dataset and in vivo experiments?
Minor Points:
- The title is very generalized, does not tell about the actual context of the manuscript. Please modify it.
- References: The references are old till 2022. Please add additional literature of 2023, 2024 and 2025.
- Conclusion: The conclusion part is too short and can be written elaboratively.
- Supplementary figure 2 is not cited in the text.
- The gene names in the entire manuscript including figures should be capitalized or either should be same.
- Line 55: the GEO code has a mistake.
